

# Research at the interface between Indigenous knowledge and soil science; weaving knowledges to understand horticultural land use in Aotearoa New Zealand

Julie Gillespie[1], Matiu Payne[2], Dione Payne[3,4], Sarah Edwards[5,6], Dyanna Jolly[6], Carol Smith[1], Jo-Anne Cavanagh[7]

[1] Department of Soil and Physical Sciences, Lincoln University, Lincoln New Zealand
[2] Kāi Tahu, Kāti Mamoe, Waitaha
[3] Māori Crown Relations, Toitū Te Whenua Land Information New Zealand, Christchurch New Zealand
[4] Vice-Chancellors Office, Lincoln University, New Zealand
[5] Landscape Policy and Governance, Manaaki Whenua Landcare Research, Lincoln New Zealand
[6] Department of Environmental Management, Lincoln University, Lincoln New Zealand
[7] Land Use and Ecosystems, Manaaki Whenua Landcare Research, Lincoln New Zealand

*Correspondence to*: Julie Gillespie (julie.gillespie@lincolnuni.ac.nz)

**Abstract.**

Addressing the complex challenges of soil and food security at international and local scales requires moving beyond the boundaries of individual disciplines and knowledge systems. The value of transdisciplinary research approaches is increasingly recognised, including those that value and incorporate Indigenous knowledge systems and holders. Using a case study at Pōhatu, Aotearoa New Zealand, this paper demonstrates the value of a transdisciplinary approach to explore past Māori food

landscapes and contribute to contemporary Māori soil health and food sovereignty aspirations. Engaging at the interface between soil science and Indigenous knowledge (*mātauraka Māori*) in an Aotearoa New Zealand context, we provide an example and guide for weaving knowledges in a transdisciplinary context. Here, *mātauraka Māori*, including *waiata* (songs) and *ingoa wāhi* (place names), provided the map of where to look and why, and soil analysis yielded insight into past cultivation, soil modification and fertilisation practices. Both knowledges were needed to interpret the findings and support

Māori to re-establish traditional horticultural practices. Furthermore, the paper extends the current literature on the numerous conceptual frameworks developed to support and guide transdisciplinary research by providing an example of how to do this type of research in an on-the-ground application.

## 1 Introduction

To address the complex challenges of soil and food security at international and local scales, there is a need for research that

moves beyond the boundaries of individual disciplines and knowledge systems (Bouma, 2015; Bouma and McBratney, 2013;



Cheik and Jouquet, 2020; Keesstra et al., 2016). Consequently, there is increasing attention to transdisciplinary research (TDR) approaches, including those that value and incorporate Indigenous knowledge systems and holders (Anthony, 2017; Chakraborty et al., 2022; Kassam, 2021; Robson-Williams et al., 2023; Smith et al., 2016). Achieving this requires challenging us, the soil science community, to see value in considering other knowledge to address these complex global challenges. There

is nevertheless a lack of empirical research that demonstrates how to connect these different knowledges within a TDR approach, particularly in relation to soils (see Gillespie et al. (2024) for discussion). To address this gap, this paper weaves together soil science and Indigenous knowledge in a TDR framework to understand food-producing landscapes in Aotearoa New Zealand. In doing so, we will provide broader methodological learnings to inform and guide soil scientists in their engagement with TDR approaches to address the needs and challenges pertaining to sustainable soil and food futures.


Soil science is recognised as naturally interdisciplinary, at the intersection of the atmosphere, biosphere, hydrosphere and lithosphere, involving a range of Western science disciplines, and, at times, aspects of the arts (Brevik et al., 2015). However, recognising the limitations of Western science disciplines, and the opportunities that engaging with other ways of knowing such as Indigenous knowledge provide (Black and Tylianakis, 2024; Stein et al., 2024), we argue that the holistic focus of

interdisciplinary research can be extended further through TDR when addressing complex global challenges. Importantly, it is the interface between society and soil that drives the shift from interdisciplinary to TDR approaches, which can be summarised as transitioning from science *for* society, to science *with* society (Scholz, 2011). This is not to reject the need for interdisciplinary research in soil science nor the narrow-focused disciplinary studies that are essential for producing knowledge of soil functions and processes (Brevik et al., 2015; Gibbons et al., 1994). Instead, TDR brings together academic disciplines,

non-academic stakeholders, and other knowledges, including Indigenous knowledge, to provide ways to understand and address these complex issues that sit beyond the capabilities of a single academic discipline (Bennich et al., 2020; Bouma, 2010; Huynh et al., 2022; Kamelarczyk and Smith-Hall, 2014; Pohl and Hirsch Hadorn, 2008; Stein et al., 2024). While there is no universally accepted definition of TDR, along with rules and guiding standards for conducting research in this manner, there are several key themes that emerge from the literature:

• TDR proves beneficial when solving complex, real-world problems encountered by society (Hirsch Hadorn et al., 2008; Lang et al., 2012; Scholz, 2011; Scholz et al., 2006).

   • TDR seeks to generate knowledge by addressing real-world problems and identifying socially robust solutions applicable in both scientific and societal contexts (Bennich et al., 2020; Gibbons et al., 1994; Knapp et al., 2019; Lang et al., 2012; Scholz, 2011; Scholz et al., 2006).

• TDR creates mutual learning opportunities between science and society, as diverse disciplines within academia, research institutions, and external stakeholders integrate their existing knowledge to produce new knowledge that is beyond any single discipline (co-production of knowledge) (Gibbons et al., 1994; Jahn et al., 2012; Knapp et al., 2019; Lang et al., 2012; Scholz et al., 2006).





- TDR is reflexive and adaptable, serving different functions to address a range of problems (Gibbons et al., 1994; Jahn et al., 2012; Lang et al., 2012). Although the absence of rules poses challenges for recognising TDR as a valid method of knowledge production, it allows TDR to respond to the dynamic and complex nature of societal challenges (Bennich et al., 2020).

- While integrating knowledge from different disciplines is important for generating new knowledge (Gibbons et al., 1994; Lang et al., 2012), TDR results in a theoretical consensus that cannot be easily reduced into its disciplinary components once established (Gibbons et al., 1994).

- TDR facilitates the development of shared conceptual and methodological frameworks, potentially diverging from existing disciplinary structures (Gibbons et al., 1994; Jahn et al., 2012; Stokols et al., 2008).

The definition of TDR offered by Lang et al. (2012) encompasses a majority of the key themes listed above: "*Transdisciplinarity is a reflexive, integrative, method-driven scientific principle aiming at the solution or transition of societal problems and concurrently related scientific problems by differentiating and integrating knowledge from various scientific and societal bodies of knowledge*" (p. 26–27).

There are several challenges to overcome when applying TDR approaches (Jahn et al., 2012; Pohl and Hirsch Hadorn, 2008). One of the main challenges is the need to preserve the integrity of the different knowledges without prioritising one over another, particularly when Indigenous knowledges are included (Kassam, 2021; Macfarlane et al., 2015; Mercer et al., 2010; Pohl and Hirsch Hadorn, 2008; Stein et al., 2024). To address these issues, many conceptual frameworks have been developed to guide TDR, (see, for example, Ball et al., 2018; Harcourt et al., 2022; Macfarlane et al., 2015; Mercer et al., 2010; Stein et al., 2024; Wilkinson et al., 2020). These frameworks are often place-based to meet the needs of the situation they intend to be used for and reflect the Indigenous knowledge they engage with, yet there are common learnings from the application of these contextualised concepts that are transferable to other place-based applications. Compared to the burgeoning literature on conceptualising TDR, there is relatively less empirical research to demonstrate how they are applied in practice. Researchers need empirical examples of TDR, that apply these conceptual frameworks in order to progress the shift in research practices. Soil science is at the edge of this transition, with strong evidence of the need to engage with TDR (Brevik et al., 2020; Friedrichsen et al., 2022; Hopmans, 2020; Rodrigo-Comino et al., 2020); however, few examples exist to guide soil scientists in this space.

To address the need for empirical soil science research engaging with a TDR approach, this paper weaves together Indigenous knowledge and soil science in a place-based case study of food production in Aotearoa New Zealand. *Mātauraka Māori[1]* is Māori knowledge, the Indigenous knowledge in Aotearoa New Zealand. It encompasses values, culture, worldviews, and

---

[1] The Kāi Tahu mita (dialect) is used in this text, where the 'ng' diagraph is replaced with 'k', e.g., mātaura**ng**a = mātaura**k**a.



philosophy relating to the environment and society (Harmsworth and Awatere, 2013; Hikuroa, 2017; Mercier, 2018; Wilkinson et al., 2020). *Mātauraka Māori* is intergenerational and place-based, codified through *pūrākau* [2] (stories), *whakataukī* (proverbs), *mōteatea* (chants), *pepeha* (quotations), *waiata* (songs), *whaikōrero* (speeches), *ingoa wāhi* (names), and *whakapapa* (genealogies) (Hikuroa, 2017; Mercier, 2018; Roskruge, 2011). The interface between *mātauraka Māori* and soil science provides an opportunity to explore connections between soil and people that move beyond the boundaries of the positivist and reductionist nature of some branches of Western science (Durie, 2004; Harmsworth, 2022; Harrison et al., 2020; Mercier, 2018). There are several examples in other research areas that demonstrate the value and importance of applying Western science alongside *mātauraka Māori* (see, for example, Forster, 2022; Harcourt et al., 2022; Harmsworth et al., 2016; Moewaka Barnes and McCreanor, 2019; Saunders et al., 2023), enabling broader and deeper understandings of the interconnections and reciprocal relations of the environment to be realised, which in turn allows appropriate and effective management approaches to be implemented. This paper contributes a soil-centred example to this body of research to support progress towards sustainable soil and food futures.

To explore the interface between soil science and *mātauraka Māori*, we have undertaken a place-based case study at Pōhatu (Flea Bay), on Te Pātaka o Rākaihautū (Banks Peninsula), situated on the east coast of Te Waipounamu (the South Island), of Aotearoa New Zealand (Fig. 1). *Mātauranga Māori*, including *waiata*, *pūrakau*, and *ingoa wahi*, indicate that *kūmara* (*Ipomoea batatas*, sweet potato) was produced at Pōhatu prior to European colonisation (Payne, 2020). With Pōhatu having been under private ownership for over 150 years, a disconnect between people and place has occurred for *whānau* and *hāpu* members, with loss of detailed knowledge of cultivation practices. This study aimed to answer the questions of *Mana Whenua* (Māori community with customary authority over the land) regarding the location of *kūmara māra* (gardens), insight into the practices used, and when the *māra kai* (food gardens) were in use by weaving together *mātauraka Māori* and soil science. The findings of this research are intended to support the reconnection of *Mana Whenua* to their land and *kai* sovereignty through re-establishing *māra kai* on this ancestral land through recovering knowledge of past land use practices, reflected by the *whakataukī* below. Beyond this context-specific application, we also provide an empirical example of how to do TDR in practice, demonstrating how to apply a conceptual framework developed to weave knowledges together to address a complex soil-centred challenge.

*Kia whakatōmuri te haere whakamua*

I walk backwards into the future with my eyes fixed on my past

---

[2] Translations are provided the first time a *kupu* (word) is used, with the Māori *kupu* used thereafter



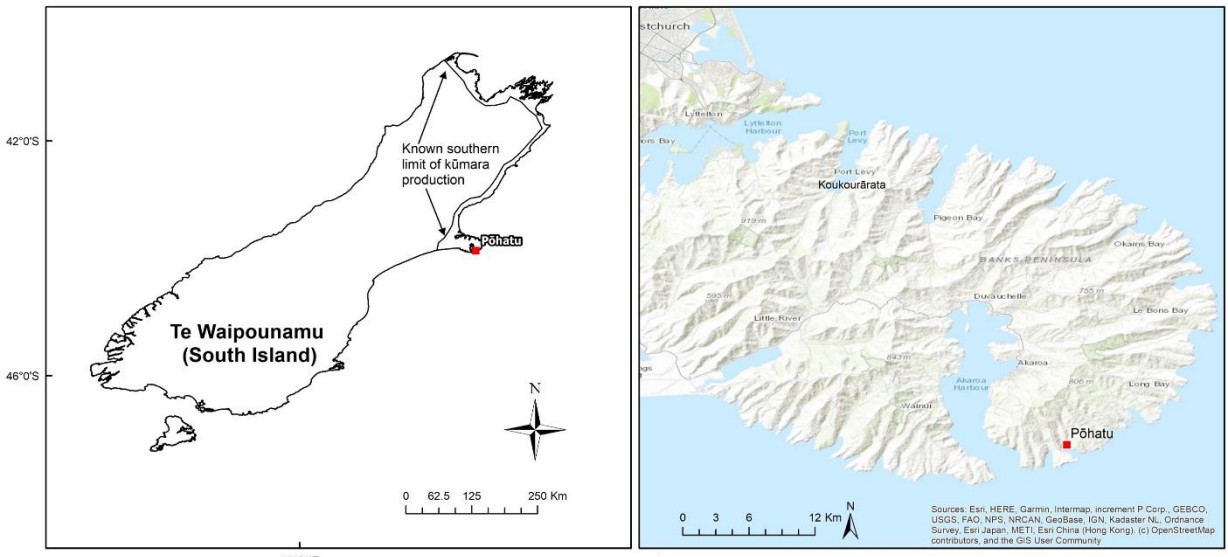

**Figure 1: Southernmost limit of kūmara production in Te Waipounamu, with location of Pōhatu study site indicated. Adapted from (Barber, 2017).**

## 1.1 Study context

Pōhatu is situated on the southeastern side of Te Pātaka o Rākaihautū (Fig. 1), for which Te Rūnaka o Koukourārata are Mana Whenua. The bay was the location of the Kāti Mamoe *pā*, Pae Karoro, translating to Pigeon Breasted, one of three Kati Mamoe *pā* sites on Te Pātaka o Rākaihautū. Kāti Mamoe occupied the Canterbury region from c.1500 CE, arriving from Te Ika o Maui (North Island). The *pā* was captured for Kāi Tahu by Tūtakahīkura, one of Moki's warriors (Taylor, 2001), in the early 1700s AD, when Kāi Tahu arrived in Te Wai Pounamu (Payne, 2020). The *pā* is situated on the hillside overlooking the beach at its

southern end and has been the primary focus of previous archaeological investigations in the bay (Brailsford, 1997; Furey, 2006; Ogilvie, 2017; Taylor, 2001). Following the 1849 Port Levy Deed of Purchase, much of the *takiwā* (area, territory) of Koukourārata, including Pōhatu, was taken against negotiation promises, with only one reserve at Koukourārata provided for Māori, resulting in *whānau* and *hapū* living throughout the bays of the tribal land leaving their homes to settle in Koukourārata (Evison, 2006). This displacement as a result of colonisation resulted in a disconnection between the people and their *whenua*

(land). European settlement in Pōhatu began with the establishment of the Flea Bay run, by brothers William and George Rhodes c. 1852 (Acland, 1946, 1951; Ogilvie, 2010), and has since been used for sheep and beef farming, as well as for conservation of environmental heritage in more recent times (Department of Prime Minister and Cabinet, 2021).

*Mātauraka Māori* associated with Pōhatu includes *pūrākau, waiata,* and *ingoa wāhi.* A Kāi Tahu *tohuka* (expert), Teone Taare

Tikao (Kāi Te Kahukura, Kāti Irakehu), discusses the symbiotic relationship between the cultivation and preparation of *kūmara* and *kauru* (a sweet food from cooked *tī kouka* (*Cordyline sp.*) trunks), in his 1870 manuscripts (Payne, 2020). In this



manuscript, Tikao details that *kūmara* beds are prepared between the different stages of *kauru* harvesting and preparation, with the *kūmara* being planted in *whitu*, the seventh month of the Kāi Tahu *māramataka* (Māori lunar calendar), corresponding to November in the Gregorian calendar (Payne, 2020). A Kāi Tahu *waiata*, Manu Tiria, further confirms this. The *waiata*,

recounted to a German missionary in 1874 at Ruapuke Island, Southland Aotearoa New Zealand, tells of how the demigod, Māui, shapeshifted into a *kererū* (*Hemiphaga novaeseelandiae*, wood pigeon) and flew to the underworld to find out who his father was. The *waiata* informs that *kūmara* is planted during the seventh and eighth months (November and December) (Payne, 2020). The significance of this is in recognising the difference in planting times between warmer northern areas and the cooler south as one of the adaptations required to successfully grow *kūmara* as what is currently recognised in the literature

as the southern-most limit of *kūmara* production (Barber, 2017; Trotter and McCulloch, 1999).

Additionally, the *ingoa wahi*, name of the bay, and its associated *pūrākau*, are particularly significant in guiding where to look and what to look for. Pōhatu translates to stone, or stony, with oral traditions from Pōhatu, and other locations where *kūmara* was grown, referring to the practice of gravel additions to garden soils to improve drainage and aid in warming (Best, 1976; Brailsford, 1997; Payne, 2020; Rigg and Bruce, 1923; Trotter and McCulloch, 1999). This name was bestowed on the *pā* in

the bay when it became the home of Tūtakahīkura (Beattie, 1990; Payne, 2020). Given the significance of *kūmara* as a crop for Māori, naming a *pā* to reflect a practice only used for *kūmara* production is a clear indicator that this crop was grown here. This name and the *pūrākau* also provide a strong indication that soils in the *kūmara māra* will contain gravel. The information from these *mātauranga Māori* sources guides where to look, and what to look for, regarding past Māori food-landscapes at Pōhatu.


Pōhatu has previously been of archaeological interest by Brailsford (1997) and Furey (2006). These authors noted the presence of two potential garden areas, Brailsford (1997) indicating a potentially terraced area at the base of the bay bordered by what appears to be a drainage ditch (Fig. 2B), and Furey (2006) a series of narrow mounds running across a north-facing slope above the present-day farmhouse (Fig. 2D).

Soils of Te Pātaka o Rākaihautū are formed from either of two parent materials. The primary rock of Te Pātaka o Rākaihautū is basalt, produced during the formation of the peninsula (Dorsey, 1988). In many places, the basaltic parent material has been mantled by loess derived from greywacke sandstone (Griffiths, 1973). A wide range of local climates are present on the peninsula due to the strong relief, with most areas below 750 m reflecting a cool temperate (montane) bioclimatic zone (Soons et al., 2002), with the eastern bays receiving around 1,000 mm/year of rainfall (Macara, 2016).




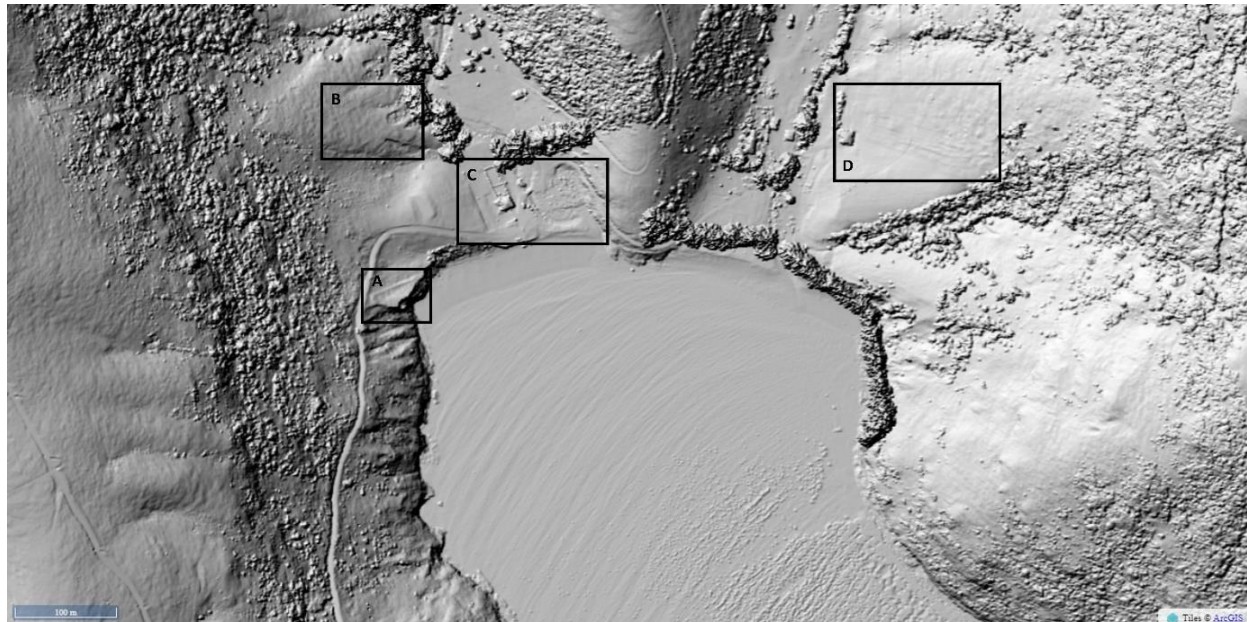

**Figure 2: Points of interest within Pōhatu. A) remains of *pā* wall, B) potential garden area surrounded by a right angled drain (Brailsford, 1997), C) living area (Brailsford, 1997). D) mounds running across the slope identified by Furey (2006).**

### 1.2 Kūmara

*Kūmara* is a root crop of importance for Māori, with both dietary and spiritual significance. Originating from South America, the crop was introduced to the Pacific Islands by transoceanic transfers, arriving in Aotearoa New Zealand during the 12th - 13th Century CE (Barber, 2012; Barber and Benham, 2024; Harburg, 2013). While across most of the Pacific *kūmara* was only

considered to be a minor crop, it was of greater importance in Aotearoa New Zealand (Barber, 2012; Furey, 2006), possibly due to the scarcity of other significant carbohydrate sources. Māori were the first to manage and crop the soils of Aotearoa New Zealand, developing extensive knowledge of the local conditions and unique soils, with over 60 *kupu* (words) describing their properties and uses (Roskruge, 2011).

A number of challenges had to be overcome to enable crop success in Aotearoa New Zealand. These include different soil types, and different cultivation methods required due to being unable to directly plant tubers in the ground across most of Aotearoa New Zealand as a result of the colder climate (Best, 1976; Yen, 1961). Māori adapted to these conditions, and *kūmara* became a staple crop. While *kūmara* was grown across most of Te Ika-a-Māui, according to writers such as Barber (2017) and Trotter and McCulloch (1999) it is limited to a coastal margin in Te Waipounamu, with Te Pātaka o Rākaihautū, and Taumutu,

at the southwestern end of Kaitorete spit, marking the southernmost extent of *kūmara* production (Fig. 1).



Plant microfossils are a useful proxy for exploring the origins and dispersal of domestic plants and the development of agriculture (Kondo et al., 1994). Within the category of microfossils, are phytoliths, starch grains, pollen, and xylem cells, which are retained in the soil (Horrocks and Rechtman, 2009). Phytoliths are particles of amorphous hydrated silica that form in the stems and leaves of plants and are deposited in the soil when a plant decays (Kondo et al., 1994). Silicic acid is taken up from the soil solution by plants, which precipitates as hydrated amorphous silica in stems and leaves (Pearsall, 2015). Phytoliths have less complex dispersal patterns than pollen grains, which can be carried by the wind (Horrocks, 2004), and are more resistant to decay than starch and pollen grains (Horrocks and Lawlor, 2006; Horrocks et al., 2002). A classification system for New Zealand was developed by Kondo et al. (1994), which separates phytolith morphotypes into different classes of grasses, trees and ferns, based on the terminology used in Japan and Europe (Carter, 2002). This has since been adopted for use in phytolith analysis in New Zealand (Carter and Lian, 2000). Additionally, the International Code for Phytolith Nomenclature (ICPN) 2.0, published in 2019 (International Committee for Phytolith Taxonomy, 2019), provides a standardised description of phytolith morphotypes to support the growing number of phytolith studies.

There have been few studies that have included the identification of *kūmara* phytoliths in Aotearoa, New Zealand. In a study of two stone mounds in Porerua (Northland, New Zealand), Horrocks et al. (2000) identified a small number of smooth, round phytoliths, most likely to have been from the pre-European *kūmara* variety, *rekamarua* based on the work of Carter (2001).

A study of a 1 ha area of modified soils at Okuora Farm on Banks Peninsula by Bassett et al. (2004), which sits adjacent to Waikākahi Pā, identified phytoliths of a spherical smooth morphotype, consistent with the phytoliths extracted from modern kūmara leaves by Carter (2001). Bassett et al. (2004) note the similarities between the phytoliths from *kūmara* and those extracted from beech, *kamahi*, *rātā* and *pōhutukawa*, which persist on the peninsula. The main difference between beech (excluding silver beech) is the verrucose surface of the phytolith (Fig. 6c), which is rough in comparison to the smooth morphotype of *kūmara* phytoliths (Bassett et al., 2004; Carter, 2001).

## 2 Methodology

To guide the TDR approach for weaving *mātauraka Māori* and soil science, we applied the He Awa Whiria (braided rivers) framework developed by Macfarlane et al. (2015). The He Awa Whiria framework identifies two streams of knowledge involved in the research (Fig. 3), each of equal value, and each with its own epistemological foundations, methods, and modes of analysis. The *mātauraka Māori* stream involved the application of *mātauraka Māori*, and *kaupapa Māori* methods and analysis. *Kaupapa Māori* refers to the methods and approaches developed and informed by *te ao Māori* (Māori worldview) and *tikaka Māori* (Māori customary protocols, procedures and rules); in other words, it refers to research done by Māori, for Māori and with Māori (Smith, 2015). The soil science stream mirrors the *mātauraka Māori* stream, comprising soil science knowledge, methods, and analysis. At times, these streams interact with each other, creating a space of mutual learning. At other times, the streams are separate, recognising that it is important to consider each of the knowledges separately to maintain





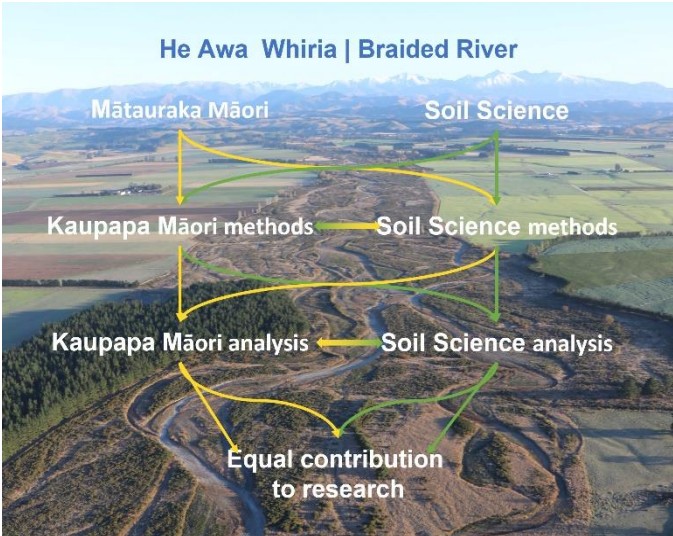

**Figure 3: The He Awa Whiria model. Adapted from Macfarlane et al. (2015) and (Wilkinson et al., 2020).**

their individual integrity (Macfarlane et al., 2015; Saunders et al., 2023). The imagery of a braided river demonstrates that while the knowledges work towards a common goal, they converge, diverge, and meander along the way (Macfarlane et al.,
220   2015).

Retaining the integrity of the knowledges involved is a critical concern of knowledge holders in both streams. Historical and ongoing interactions of Western science with Indigenous knowledge have been extractive, treating Indigenous knowledge and people as research subjects (Saunders et al., 2023; Smith, 2021). Conversely, there is an enduring concern within the scientific
community that prioritising societal concerns may introduce individuals who lack the required expertise into the knowledge production process, leading to a loss of scientific quality (Bouma, 2015). Saunders et al. (2023) discuss the tensions between researchers, knowledge holders, and stakeholders, who stress the importance of finding an 'appropriate balance' between producing new knowledge and recognising who this knowledge is for. The He Awa Whiria framework addresses these concerns by acknowledging the importance of respecting and retaining the integrity of both knowledge streams. It allows for
flexibility, ensuring that the knowledge systems can operate independently when appropriate (Macfarlane et al., 2015; Saha et al., 2023; Wilkinson et al., 2020).

This framework is also accompanied by the *whakataukī*, "*nā tō rourou, nā taku rourou, ka ora ai te iwi; with your food basket and my food basket, the people will prosper*". This *whakataukī* speaks to the foundations of this research, bringing together
the two different knowledge systems to answer the questions of Mana Whenua more comprehensively than either of the knowledges alone are able to do in this context (Macfarlane et al., 2015; Wilkinson et al., 2020).



The He Awa Whiria framework is suitable for research teams with any ratio of Māori to non-Māori due to the dynamic nature of the knowledge streams converging and diverging throughout the research process (Wilkinson et al., 2020). Nevertheless, a

level of humility is required throughout research guided by this framework, particularly by non-Māori researchers trained in Western science disciplines that have a privileged status in relation to *mātauraka Māori* (Smith, 2021). We were mindful of these power dynamics as we embarked on this research as a team of *Pākehā* (New Zealander of European descent, JG), Māori (MP, DP), *Tauiwi* (settler/recent migrant, SE, DJ, CS, JC). While JG was the overall project leader, she was guided through the research by DP who also enabled the connection with *Mana Whenua* at Te Rūnaka o Koukourārata, including the key

holder of *mātauraka Māori* (MP). We provide details of the engagement with *Mana Whenua* and *mātauraka Māori* in the following section.

## 3 Methods

### 3.1 Engagement with Mana Whenua

A crucial first step for this research was to build a relationship with Mana Whenua and learn about the issues they faced on

their whenua, including the aspirations they held for future land use. This began with *whakawhanaungatanga* (relationship building), which predominantly took place at Koukourārata from October 2020, where JG engaged with *whānau* to support a *māra kai* project in Koukourarata (Port Levy), which involved planting a range of Māori and non-Māori potatoes for the community. The leader of this project was MP, the then Chair of the *Rūnaka* and key holder *mātauranka Māori* and *māra kai*. As such, this relationship was already established when the opportunity to undertake this study at Pōhatu was identified. One

of the areas of interest for MP, was revitalising the planting and harvesting of *kūmara* in the Koukourarata *takiwā* (area) and recalled the *pūrakau* of Pōhatu Pā and *waiata* referencing the planting time in Te Waipounamu. He also mentioned the methods of warming the soil to combat the colder climate and was keen to discover whether those stories could be validated. Utilising that information, a project was co-developed, and permissions were gained from the current landowners and MP as *Mana Whenua* for this research to be carried out. As additional support, DP, who was Deputy Vice–Chancellor, Māori and Pasifika,

and leader of *Mataurака Māori* research at Te Whare Wānaka o Aoraki Lincoln University, joined the research project as a co-supervisor.

A preliminary trip to Pōhatu occurred for the primary focus areas to be determined in April 2022. Soil sampling occurred in May 2022 (see Section 3.2), and soils were analysed by JG (see Section 3.3). During the analysis period [May 2022 – July

2023], initial findings were discussed by JG, MP and DP, with a physical resource produced to support these discussions and the interpretation of the findings. This was in the form of a folder and included maps, Digital Elevation Models, graphs, photographs and descriptions, and was left with *Mana Whenua* for the findings to be readily shared with other *whānau* members. During these discussions, a number of questions that were raised in the interpretations when applying a soil science lens were addressed, and further pathways for analysis and evaluation were identified. These were then pursued with further



work, and the overall findings were presented to *Mana Whenua*, with ongoing discussion occurring, demonstrating the weaving of knowledges throughout the research process. Key interactions are described alongside the findings in Section 4, with a detailed discussion of the process of weaving knowledges provided in Section 5.

## 3.2 Site selection and sampling

An initial auger survey across the slopes identified by (Brailsford, 1997) and (Furey, 2006) identified buried soil horizons and small greywacke sandstone gravels in area D identified by Furey (2006), while evidence of māra was not clear on slope B. Therefore, in this paper, we investigate the slope with mounds identified in Fig. 2D. Transects were augured to a depth of 1 m across the hillslope identified in Fig. 2D, perpendicular to the raised earth lines, both on and between the mounds. This identified that part of the slope had been influenced by a landslide with soil material of a lighter colour, finer texture, and

lower permeability, burying darker soil horizons. From this, two pit locations were identified, one on the non-landslide part of the slope and the other on the part that had been disturbed by the landslide (Fig. 4). Pits were then hand dug to a depth of 100 cm, described following Milne et al. (1995) and classified according to the New Zealand Soil Classification (Hewitt, 2013). The profile was sampled by horizon to a depth of 100 cm for soil and phytolith analyses. Samples were air-dried and sieved to 2 mm in preparation for soil analysis.

## 3.3 Soil analysis

### 3.3.1 Microfossil extraction and classification

Extraction of phytoliths from soil samples were based on the methods of Carter (2001) and Parr et al. (2001). A minimum of 300 counts were performed for each slide under plane polarised light at 400 x magnification (Olympus BX53-P), requiring 5

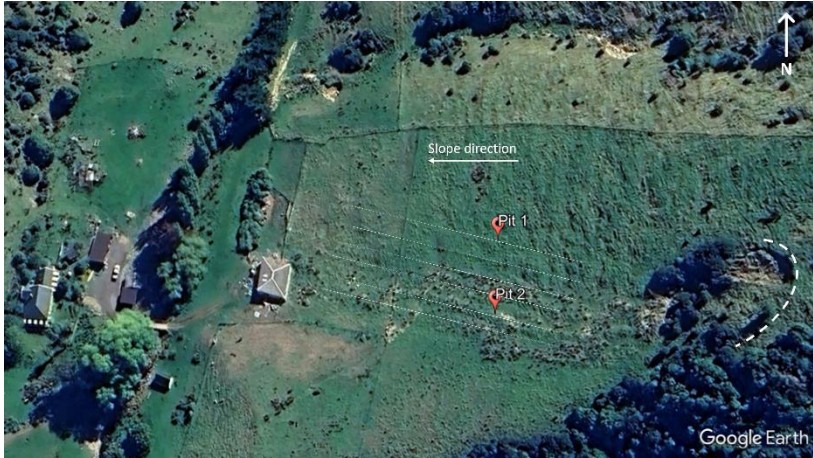

**Figure 4: Location of soil pits. Raised mound positions are indicated by straight dashed lines. The origin of the landslide is indicated to the right of the image by the curved line. © Google Earth.**





– 15 fields of view. Phytoliths were classified into one of 12 different phytolith morphotypes, including an 'other' category

for unidentifiable forms.

Phytoliths were described using Kondo et al. (1994) to draw comparisons with previous New Zealand research. Correlation to the ICPN 2.0 (International Committee for Phytolith Taxonomy, 2019) has also been provided to align with universally accepted nomenclature and classification. The work of Carter (2001) and Bassett et al. (2004) was used in identifying potential kūmara phytoliths, alongside extractions of fresh plant material. Diatoms were also counted, but taxonomic identifications

were not made.

Starch grain extraction followed the methods of (Horrocks, 2005), and grains were counted under cross-polarized light at 100 x magnification (Olympus BX53-P). The full area of the coverslip (22 x 22 mm) was systematically examined, and the total number of starch grains were counted. A classification system for starch, as exists for phytoliths, is absent, with starch grains identified by comparisons to reference collections (Arráiz et al., 2016).

Charcoal extraction followed the method of Rhodes (1998), where 2.0 g samples were digested with 6% $H_2O_2$ and counted in a petri dish under a stereoscope (Nikon SMZ645) with top lighting at 40 X magnification. Samples were moistened prior to observation to increase their lustre and aid in differentiating the charcoal from other dark organic fragments in the sample. Samples were systematically scanned and the total number of fragments in the petri dish were counted.

Analysis of microfossil counts used the tidypaleo package in R (Dunnington et al., 2022), where phytolith counts were

converted to percentages and displayed as frequency diagrams, with starch and charcoal counts presented.

### 3.3.2 Soil signatures

Soil samples from soil horizons with evidence of modification were radiocarbon dated at the University of Waikato using Accelerator Mass Spectrometry. Stable Isotope analysis for $\delta^{15}N$ and $\delta^{13}C$ was completed at Lincoln University by elemental analyser continuous flow isotope ratio mass spectrometry (EA-CF/IRMS; Sercon GSL/ 20-22). ICP-OES was used to

determine the trace element concentrations of soil and plant, with samples prepared using nitric acid microwave digestion to analyse for As, Ca, Cd, Cr, Cu, K, Mg, Mn, Mo, Ni, Pb, Zn (Agilent ICP-OES5110).

### 4 Findings and interpretation

### 4.1 Soil profile

Six earth rows, angled at approximately 45° to the slope, were identified. The initial auger survey identified that three of these

mounds appear to have been influenced by the landslide indicated in Fig. 4, with a yellowish-brown horizon (10 YR 5/4) present between the topsoil and a very dark brown (10 YR 2/2) underlying a buried A horizon containing fine to medium greywacke sandstone beach gravels. One pit was dug in the area not influenced by the landslide (Pit 1), and the second (Pit 2) on a mound within the landslide influenced area (Fig. 4, Fig. 5). Soil profile descriptions are presented in Tables 1 and 2. An absence of large charcoal or woody remains was noted in the profiles, including in the modified horizons. Clay content




increased with depth, and there were no observed increases in the sand content within any horizon, with particular attention paid to those that were modified. Evidence of modification included the presence of fine to medium (20–60 mm) rounded greywacke sandstone gravels, which contrast with the autochthonous angular basalt clasts (20–100 mm) in the unmodified soil horizons.

**Table 1: Soil profile description of Pit 1, described according to Milne et al. (1995) and classified as a Typic Orthic Melanic Soil (Hewitt, 2010) (Haplustepts.(Soil Survey Staff, 1999)).**

| Horizon | Colour | Texture | Structure | Coarse fragments | Size mm (%abundance) |
|---|---|---|---|---|---|
| Ap | 10 YR 2/2 | ZL | Strong, fine, polyhedral | Rounded greywacke sandstone (GWSS) | 20 - 60 (5%) |
| A/B | 10 YR 2/2 10 YR 5/3 | ZL | Strong, fine–medium, polyhedral | Rounded GWSS | 20 – 60 (5%) |
| Bw(f)1 | 10 YR 5/3 (7.5 YR 5/6) | ZL | Moderate, fine–coarse polyhedral | Rounded GWSS | 20 – 60 (5%) |
| Bw(f)2 | 10 YR 5/3 (7.5 YR 4/6) | ZC | Weak, medium, blocky | Angular basalt | 20 – 100 (3%) |

**Table 2: Soil profile description for Pit 2. The soil has been described according to Milne et al. (1995) and classified as a Mottled Mafic Melanic Soil (Hewitt, 2010) (Udolls or Haplustepts (Soil Survey Staff, 1999)).**

| Horizon | Colour | Texture | Structure | Coarse fragments | Size mm (% abundance) |
|---|---|---|---|---|---|
| Ap | 10 YR 3/2 | ZL | Strong, very fine – fine, polyhedral | Angular basalt | 6 – 10 (5%) |
| 2Ap | Darker than 10 YR 2/1 | ZL | Strong, very fine – fine, polyhedral | Rounded GWSS | 6 – 10 (5%) |
| Bw(g) | 10 YR 5/4 | ZL | Moderate, very fine–medium polyhedral | Angular basalt | 6 – 60 (8%) |
| 3bAp | 10 YR 4/3 | ZL | Weak, very fine–medium, polyhedral | Rounded GWSS | 2 – 20 (5%) |
| 3bBw1 | 10 YR 5/4 | CL | Weak, fine–medium, polyhedral | Angular basalt | 6 – 60 (10%) |
| 3bBw2 | 10 YR 5/4 | CL | Weak, very fine – fine, polyhedral | Angular basalt | 6 – 60 (10%) |




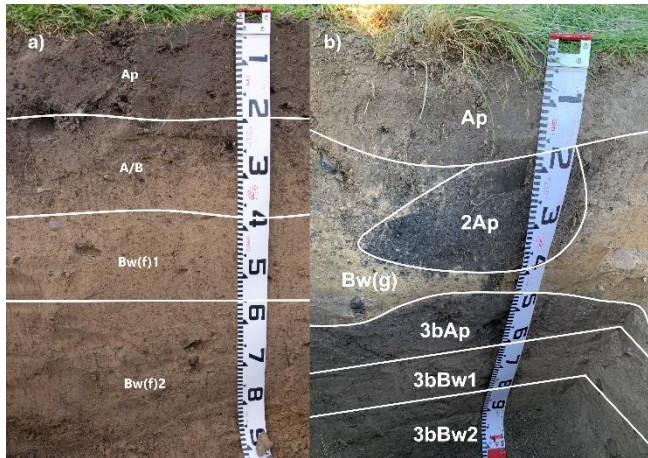

**Figure 5: a) Pit 1, without landslide disturbance. Arrows indicate beach gravel additions to the soil. b) Landslide influenced soil (Bw(g) is the landslide material), covering the original soil surface (bAp), with modification present in the Ap2). Beach gravels were present in the Ap2 and 3bAp horizons.**

*Pūrakau* associated with the chief of Pōhatu Pā, Tūtakahīkura, identifies that the *pā* was named to acknowledge the gravel added to the soil to increase the soil temperature and improve drainage, creating more suitable growing conditions for *kūmara* (Payne, 2020). As this practice was only used for *kūmara* and considered wholly unnecessary for other crops (Trotter and McCulloch, 1999), the presence of gravel in the modified soil horizons is a clear indicator that *kūmara* was grown at Pōhatu.

Analysis of soils on the slope with raised earth lines, identified by Furey (2006), revealed the presence of small, rounded greywacke sandstone gravels in horizons with darker soil colourings, contrasting with the autochthonous angular basalt clasts present in the remainder of the profile. Owing to the elevation and aspect of this slope, it is not possible for these rounded gravels to have been emplaced in these soils by any surface processes or other agents of transport. An increase in sand content is a feature reported in modified soil horizons to lighten the soil (McFadgen, 1980a), which is noticeably absent in the modified

horizons of the Pōhatu soils. This could be due to the strongly structured nature of the Melanic Soil material allowing for drainage and the additions of gravel being sufficient in retaining temperature for *kūmara* production.

Alongside the additions of beach gravels to the soil, properties of the soil profile indicate modification for *kūmara* production, particularly in Pit 2. Due to the continued weathering and pedological development of the soil in Pit 1, which has not been

influenced by a landslide, modifications are not as obvious as those in Pit 2. In Pit 2, the pocket of dark 2Ap soil material developed into the Bw(g) landslide deposit (Fig. 5B) resembles *kūmara* pits in other Te Waipounamu locations throughout Aotearoa New Zealand that have been identified by Barber and Higham (2021) and Gumbley et al. (2004). In other locations, particularly in Te Ika-a-Māui, *kūmara māra* consists of individual mounds, termed *puke* (Best, 1976; Furey, 2006; Gumbley et al., 2004; Walsh, 1902), but do not appear in archaeological investigations in Waitaha (including Bassett et al., 2004; Furey,





2006; Jacomb, 2000; Morris, 1994). These individual mounds are not present at Pōhatu; however, the pocket of enriched material in the 2Ap horizon of Pit 2 indicates that a similar planting arrangement may have been used within the continuous rows. Enriched pockets in elongated mounds have not been reported elsewhere and are likely to have arisen from the need to recommence *kūmara* production in the poorer quality landslide material. The use of elongated mounds does not appear in the literature beyond sites identified on Te Pātaka o Rākaihautū (Brailsford, 1997; Furey, 2006; Jacomb, 2000; Morris, 1994).

Being located at the southern limits of *kūmara* production, reducing the exposed surface area of individual *puke* may have served the purpose of retaining soil temperature while still raising the plants in mounds as is common practice for *kūmara* (Best, 1976; Law, 1969; McFadgen, 1980b; Trotter and McCulloch, 1997; Walsh, 1902; Yen, 1961).

Previous archaeological investigations at Pōhatu have not looked below the surface, which has left questions as to the
composition of the raised mounds identified by Furey (2006). Similar surface features at *māra* sites on Te Pātaka o Rākaihautū have been identified as stone rows (Bassett et al., 2004; Furey, 2006; Harrowfield, 1969; Jacomb, 2000; Morris, 1994), defined by Walton (1999) as elongated heaps of stone. No stone rows were identified at Pōhatu, with our investigation identifying that the mounds presented in Furey (2006) are earthen.

**4.2 Microfossil analysis**

Phytolith counts of 300 phytoliths per slide, requiring 5 – 15 fields of view, were completed for each horizon in the two soil pits. Elevated amounts of phytoliths with a spherical smooth morphotype (Fig. 6a, b), consistent with phytoliths extracted from modern *kūmara* leaves and previous phytolith research (Bassett et al., 2004; Carter, 2001; Horrocks and Rechtman, 2009; Horrocks et al., 2000), were observed in the modified soil horizons of both pits. Notably, these phytoliths displayed a line of
grooves at the hemisphere, absent in other phytoliths with spherical smooth morphotypes from native tree species, including beech, *kamahi*, *rātā*, and *pōhutakawa* (Bassett et al., 2004). Furthermore, starch grains (Fig. 7) were present in and below modified soil horizons (Fig. 8). This supports the *mātauraka Māori* of *kūmara* production at Pōhatu.



The elevations of spherical smooth phytoliths corresponded to an overall decrease in phytoliths with spherical verrucose and

point-shape (arrow) morphotypes from the 3bBw3 of Pit 2 through to the present-day topsoil, indicating that more trees and

tussock grasses were present in this environment prior to soil modification 9Fig. 8). An increase in grass phytoliths with an

elongate smooth morphotype occurred in the Bw(f)1 soil horizons of Pit 1, where the beach gravels were located. This increase

is also observed in the 2Ap horizon of Pit 2, becoming more abundant above the 3bAp, from the time of the landslide. In Pit

2, increases in platy jigsaw phytoliths, associated with ferns, and diatoms were observed in the modified horizons, indicating

that ferns and marine materials may have been soil amendments to develop the growing beds. An increase in phytoliths

associated with warmer climate grasses was present in the Bw(g) of Pit 2, with fewer identified in the modified 2Ap, before

increasing again in the modern-day topsoil. A spike in truncated cone chionochloid phytoliths is visible in the 3bAp horizon,

indicating plants such as *toetoe* (*Austrideria spp.*) may have been added in reasonable volumes to enrich this horizon. *Toetoe*

are abundant across Aotearoa New Zealand, found in almost all environments, with many uses by Māori, including as a

building material, bedding, wound treatment, weaving, and medicinal purposes (Hiroa, 1949; Scheele and Sweetapple, n.d.;

Tipa, 2012); however, there is no record of it being used as a soil amendment.

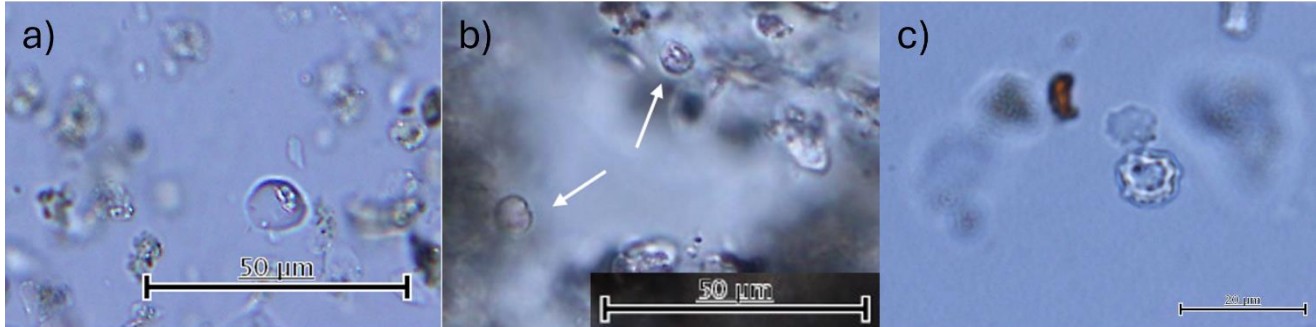

**Figure 6: a) Spherical smooth phytoliths extracted from Pōhatu soil sample 2Ap of WS13. b) Spherical smooth phytoliths isolated from modern *kūmara* leaf. c) Spherical verrucose phytolith from Pōhatu soil sample. Scale bar = 50 µm for a) and b), and 25µm for c).**

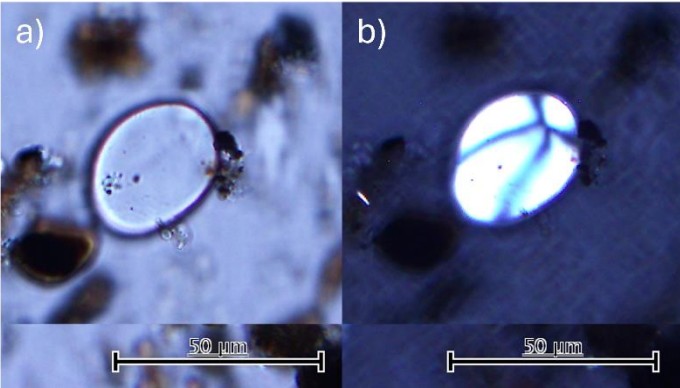

**Figure 7: a) Starch grain extracted from the 2Ap horizon of Pit 2. b) viewed under cross-polarised light, showing the Maltese cross, characteristic of *kūmara* grains. Scale bar = 50µm**





**Figure 8: Stratigraphic phytolith and starch count plots for Pits 1 and 2.**



While no large charcoal fragments were visible in the soil profiles, charcoal fragments (<500 µm) were identified in considerable amounts in the modified soil horizons, with over 300 fragments counted per 2.0 g unit of soil from the 2Ap horizon of Pit 2 (Fig. 8). This increase, and a smaller spike in the 3bAp horizon of Pit 2, indicates potential ash additions to enhance the soil, or the burning of in situ vegetation to clear the land between growing periods. The comparison between Pits 1 and 2 indicates that both are likely to have occurred, with a higher concentration of charcoal in the modified horizons than the horizons containing gravels in Pit 1.

## 4.4 Soil amendments

Trace element analysis identified an increase of manganese (Mn) concentrations in the modified horizons of Pit 2 (Fig. 9), while the other trace elements analysed showed little variability across the different horizons. The increase in Mn was considered to potentially arise from soil amendments to improve kumara growth. Initially, seaweed and penguin guano were considered as potential fertiliser sources based on the work of Morris (1994). An examination of *māra* soils at Panau and Kirikiriwaerea (Menzies Bay), nearby bays on Te Pātaka o Rākaihautū, investigated the influence of penguin guano and seaweed fertiliser additions as soil amendments (Morris, 1994). This study noted that soils with penguin influence were dark-coloured, particularly in pale-coloured loess-derived soils; identifying that Mg, Ca, and K concentrations were lower in soils influenced by penguins. Due to manure of all kinds considered *tapu* (sacred, prohibited, restricted) by Māori (Best, 1976), the continuing addition of guano to the soil is perhaps an unlikely practice despite its potential availability at Pōhatu as a *kororā* (little blue penguin, *Eudyptula minor*) colony. *Mātauraka Māori* for Kirikiriwaerea identify seaweed as an important fertiliser for *māra* soils (Morris, 1994), which leads to elevations in As, B, Mn, Cd and a decrease in Mo (Blanz et al., 2019). Increases

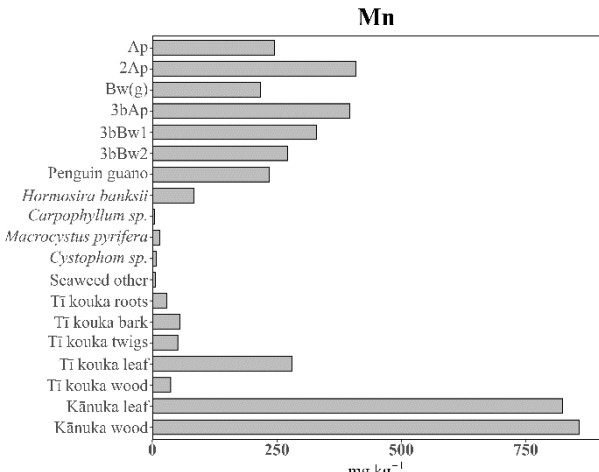

**Figure 9: Mn concentrations in Pit 2 soil horizons and potential inputs. The 2Ap and 3bAp horizons are modified. Seaweeds =**
*Hormosira banksii, Carpophyllum sp., Macrocystus pyrifera, Cystophom sp.,* **Seaweed other.**



of these trace element metals were also not observed at Pōhatu, where Mn was elevated in both modified horizons, and Mo increased in the 2Ap horizon. These trends were not observed in the modified horizons at Pōhatu, where an increase in Mn was the key observation for the modified horizons, with lower levels of Mg, Ca, and K than reported for Panau and Kirikiriwaerea.

All Pōhatu seaweed samples were high in As, Ca, K and Mg, while the *rimurimu* (*Macrocystis pyrifera*, giant kelp) species was also high in Cd and Cr, and Cystophora was high in Mo. However, seaweed and guano did not correspond to elevations of this trace element in the modified soil horizons. The increase of Mn in the modified soil horizons of Pit 2 exceeds the concentrations found in the penguin guano and seaweed samples analysed, suggesting that these are not nutrient sources in the Pōhatu *māra* soils. This finding led to discussions with *Mana Whenua* to identify other potential fertiliser sources. In their paper on *mātauraka Māori* technologies, Payne (2020) discusses the interconnectedness between the practices of *kūmara* cultivation and *mahika kauru*, an important carbohydrate source extracted from *tī kouka*. This prompted an expansion of the trace element micronutrient and stable isotope analyses to include different parts of the *tī kouka* tree.

Further literature review also highlighted the use of *mānuka* (*Leptospermum scoparium*; tea-tree) and *kānuka* (*Kunzea ericoides*; white tea tree) as palisades around garden areas and as fast-establishing plants during fallow periods, which were burnt in situ to clear the *māra* for the next planting season (Rigg and Bruce, 1923; Walsh, 1902). Rigg and Bruce (1923) note the enduring fertility of the modified Māori soils with *mānuka* ash additions on the Waimea Plains, which were highly sought after by European growers. Analysis by Miller et al. (1955) of soils (not Māori modified soils) where a stand of *mānuka* was burnt concluded that the addition of ash resulted in nutrient levels comparable to those released to soils through the application of common fertilisers. As the more abundant of these two species in the study area, known particularly for its rapid re-establishment following fire (Harris and Harris, 1939; Wilson, 1994), *kānuka* were also analysed.

The concentration of Mn in the *kānuka* samples was higher than in the modified soil horizons, suggesting *kānuka* as a likely fertiliser source. *Tī kouka* bark and leaves showed elevations of Ca, Cr, Cu, K, and Zn, however, these did not correspond to the trace element concentrations present in the soil and did not strongly indicate it as a likely soil addition. Neither *kānuka* nor *tī kouka* produce phytoliths (Horrocks, 2004; Kondo et al., 1994), thus cannot be cross-checked through this method. An analysis of the soil pollen record may be a potential option to investigate this further; however, pollen recovery following prolonged heat/fire exposure is poor (Ghosh et al., 2006). Heat and fire destroy pollen grains, and due to the long cooking time for *tī kouka* and depending on the burning practices for *kānuka*, pollen grains are unlikely to be present even if parts of these trees were used as fertilisers.

Further soil enhancement may have stemmed from experience in working with the different soils that occur at Pōhatu, capitalising on the geochemical properties of the soils. Māori have extensive knowledge of soil properties and capabilities,





with over 60 terms for soils (Harmsworth and Roskruge, 2014; Roskruge, 2020). As the second of three shields that form the Te Pātaka o Rākaihautū volcanic complex, the primary parent material of the peninsula is mildly alkaline basalt (Stipp and McDougall, 1968). These rocks are high in Mg and Fe oxides, released during weathering to form strongly structured Melanic

Soils (Haplusteps, Udolls (Soil Survey Staff, 1999)) with high fertility (Hewitt, 2010; Price and Taylor, 1980). The second parent material of the soils in the study area is loess, blown across the Canterbury Plains during interglacial periods, and mantles parts of the peninsula, forming Pallic Soils (Fragidalfs, Fragiochrepts, Haplustepts (Soil Survey Staff, 1999)). These deposits are predominantly of a silty texture, produced from greywacke sandstone containing quartz and plagioclase feldspars ($NaCaSi_3O_8$ to $CaAl_2Si_2O_8$) as the dominant mineralogy (Griffiths, 1973; Raeside, 1964). This quartzofeldspathic material

results in Pallic Soils that are low in free iron and ferromagnesian minerals (Hewitt, 2010; Raeside, 1964). These soils have low permeability and lower fertility than those formed from basalt.

Within Pit 2, soils from both of these parent materials are present. Soils of basaltic origin form the original soil profile, and the paler soil material of loess origin comprises the landslide layer. This landslide layer (Bw(g)) had a lower Mn concentration

than the original subsoil horizons (Fig. 9), derived from basaltic parent material. The Mn concentration in both modified horizons of Pit 2 exceeds that of either soil of the two parent materials, indicating that organic additions that are high in Mn have been used to enrich the soil. The highest concentration of Mn is in the 2Ap horizon, developed into the loess soil deposit. Mixing Melanic Soil material, along with the organic additions previously discussed, may have contributed to this high concentration. These naturally richer soils may have been recognised as superior for *kūmara* based on experience within the

bay and sought this as a base to develop a highly fertile soil. This potentially came from recognising that the Pallic Soil material is poor for growing and the Melanic Soil material is more suitable for *kūmara* production. The properties of the Melanic Soils of Pōhatu align with the description of *kirikiri tuatara*, a brown, friable and fertile soil that is suited to *kūmara* production, while the landslide material is better described as *onetuatara*, a stiff brown soil needing amendment with sand or gravel to suit *kūmara* (Roskruge, 2020).


In addition to soil and plant trace element concentrations, stable isotopes $\delta^{13}C$ and $\delta^{15}N$ have been used to identify potential sources of enrichment for the modified māra soils. A slight enrichment of $\delta^{13}C$ (less negative values) in the 2Ap horizon (extensively modified) of the slope with raised earth lines indicates a different input source than those contributing to other soil horizons, with potential additions of C4 terrestrial plants or marine sources. As the only known C4 plants to become

naturalised in Aotearoa New Zealand are pastoral grasses (Crush and Rowarth, 2007), with no known native or endemic species, it is improbable that these were added, with seaweeds or other marine additions being more plausible. The main C4 species include maise, sugar cane and sorghum, which were absent in Aotearoa New Zealand until the arrival of Europeans. All soil samples fall within the $\delta^{13}C$ range of terrestrial C3 plants (-34 - -22‰) identified by Hawke and Clark (2010). Some seaweeds, including *Macrocystis pyrifera* (giant kelp), have been reported to follow photosynthesis pathways that are more

aligned to C4 plants (Yanhui and Zhigang, 2016), which results in less negative $\delta^{13}C$ values. All soil $\delta^{13}C$ values (-28.49 – -



25.13) are within the expected range of influence by C3 terrestrial plants (-34 – -22‰ (Hawke and Clark, 2010)). A less negative δ13C value in the 2Ap horizon of Pit 2 indicates that small amounts of marine sourced nutrients, such as seaweeds (-20.6 and -13.3‰), have potentially been added to the soil. Penguin guano (-25.91 – -27.07‰) was more negative than expected based on the findings of (Hawke and Clark, 2010) (Fig. 10).


The peak of δ13C in the 2Ap horizon reflects the findings of other studies where an increase in organic matter has been observed (Wilson et al., 2007). In forested canopies, when leaves are incorporated into topsoil, fresh leaf litter accumulations retain the more negative δ13C values, which are then slowly degraded by bacteria (and isotopically fractioned) towards less negative δ13C values (Rogers et al., 2017). Marine sources also reflect a less negative δ13C (Kinaston et al., 2013), as demonstrated in the 485 analysed seaweed samples. It is possible that small amounts of seaweed were added to develop the 2Ap horizon, but the ash of C3 plants were likely the primary addition.

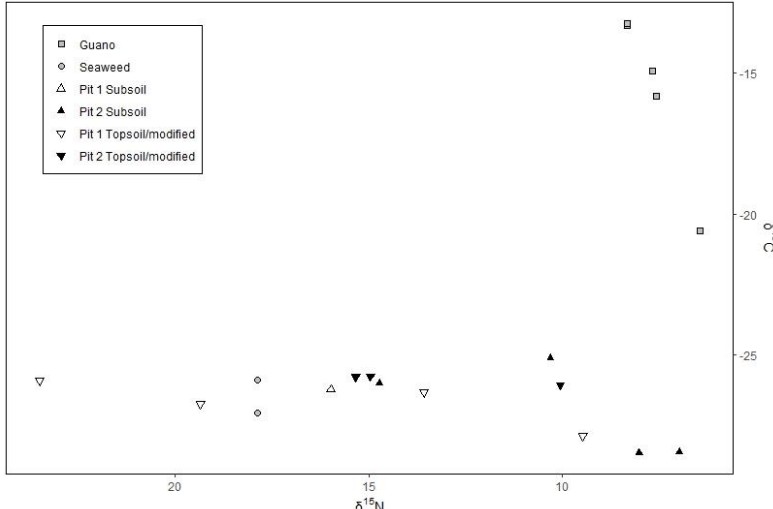

**Figure 10: Stable isotope ratio for Pits 1 and 2, with potential organic additions of seaweed and penguin guano**

### 4.5 Radiocarbon dates

*Mātauraka Māori* for Pōhatu is associated with Kāi Tāhu occupation. *Mātauraka Māori* informs us that Kāi Tāhu arrived c.
1620 AD (Payne, 2020). However, the mātauraka Māori does not identify if *māra* were present upon arrival. If *māra* were present at this point, it would indicate previous cultivation by Kāti Māmoe, or if Kāi Tahu initiated *kūmara* production at Pōhatu. This uncertainty raises the question: did the name Pōhatu, referring to the additions of gravel to the soil, arise from what was observed upon arrival, or was the name decided after this practice began for Kāi Tāhu planting? Other studies recognise that Kāti Māmoe were growing *kūmara* in other areas within Waitaha (Law, 1969), making it plausible that *kūmara*
was being grown at Pōhatu prior to the arrival of Kāi Tāhu.



To determine when the *māra* were in use, radiocarbon dating was conducted on the modified soil horizons in Pits 1 and 2 on the slope with earth lines (Fig. 2D). The dates obtained for the 2Ap horizon of Pit 2, 279 ± 19 (1680 AD), align with Kāi Tāhu occupation. The radiocarbon date obtained for the original *māra* layer, b3Ap of Pit 2, was 662 ± 16 (1400 AD), predating the arrival of Kāti Māmoe in Te Wai Pounamu, which occurred c. 1500 AD (Beattie, 1990). The date of the soil material from where the rounded gravels were in the greatest abundance for Pit 1 was 1175 ± 17 (370 AD), predating any known settlement in Aotearoa New Zealand. An absence of large pieces of charcoal or woody material in the modified soil layers resulted in radiocarbon dating being obtained from bulk soil samples for each of the modified horizons, overestimating the dates of the soils, indicating them as older than their most recent organic inputs.

The bulk sample analysis of radiocarbon by AMS incorporates different pools and fractions of C within the soil, which have different stabilisation mechanisms, resulting in obscured information on the distribution of the radiocarbon age of the soil organic matter (Rosenheim et al., 2013; Stoner et al., 2022). Consequently, the radiocarbon ages obtained are likely to be older than when these soils were used as *māra*, as the date accounts for both old, recalcitrant C and young, more labile C added to the soil as organic matter. This issue is particularly relevant for the gravel-modified layer in Pit 1 and the 3bAp horizon in Pit 2, where the radiocarbon dates do not appear to accurately reflect the most recent additions of C to the soil, making it difficult to determine when the māra was in use. Based on the results of the radiocarbon dating analysis and *mātauraka Māori*, it remains unclear whether the māra were in use by Kāti Māmoe before the arrival of Kāi Tāhu in the early 1700's. The radiocarbon date determined for the 2Ap horizon of Pit 2 is likely more accurate due to the large amounts of microscopic charcoal in the soil.

## 5 Discussion

This case study has explored the interface between *mātauraka Māori* and soil science in a place-based context. The He Awa Whiria framework was critical in conceptualising and guiding the weaving of knowledges in this research and ensuring meaningful engagement throughout the research process. This approach allowed both *mātauraka Māori* and soil science to contribute equally alongside each other, ensuring that the research is holistic and culturally informed and enabling us to answer the research questions more comprehensively than either knowledge alone could have.

Our research demonstrates the value of engaging with knowledge outside of the Western science paradigm through genuine and non-extractive interactions with *mātauraka Māori*. While recognising *mātauraka Māori* as a knowledge system of equal value to Western science, it has also been possible to maintain scientific integrity. This demonstrates that there is space for both knowledges, with each having a distinct role and contribution. *Mātauraka Māori* has guided and informed the research questions and context while soil science methods are applied to conduct the soil analysis. The scientific rigour of the study is





maintained through applying conventional soil science methods. Yet the significance of the findings, such as confirming where *kūmara* was produced and identifying the horticultural practices used, is validated through the acceptance and interpretation

of the findings by *Mana Whenua,* concluding that the findings support *mātauraka Māori* for *kūmara* production at Pōhatu. This approach respects both the scientific and cultural dimensions of the research, ensuring that neither knowledge stream is subsumed by the other.

Of equal importance to retaining the individual integrity of the knowledges is facilitating and utilising mutual learning

opportunities (Macfarlane et al., 2015). These opportunities arose when the knowledge streams of He Awa Whiria intersected (Cram et al., 2018; Macfarlane and Macfarlane, 2018; Wilkinson et al., 2020). Discussions around the soil science findings, their alignment with existing knowledge, and how they fit within broader understandings of the relationships between soil, food and people in this landscape contributed to developing a more complete picture of past land use at Pōhatu. In instances where soil science results were inconclusive, *mātauraka Māori* provided guidance on what to consider next.


To facilitate discussion and interpret findings, a folder of information was prepared and provided to *Mana Whenua*. This folder served as a connector, aiding communication of the scientific results. It included maps, images, descriptions of all potential sites within the bay, annotated photographs of the soil profiles, phytolith stratigraphs, graphs of micronutrient concentrations, stickers with questions, suggested interpretations from a soil science lens, diagrams, and additional observations beyond the

primary focus area of potential interest to *Mana Whenua*. This format enabled easy reference during discussions.

A timeline of occupation and activities at Pōhatu was also part of the resource, which included dates from known oral histories and traditions and findings from this study. The timeline became a focal point of discussion, with further detail added during the discussion, including names and dates of notable movements and events, providing further context and supporting

interpretation of the original research questions. The folder was a key resource for engaging the two knowledge streams within the mutual learning area of the He Awa Whiria framework.

The interaction between the knowledges in this case study provided an opportunity to develop a more complete understanding of past horticultural land use and support the rediscovery of *māturaka Māori*, which neither knowledge alone would have been

able to achieve. With *mātauraka Māori* having been lost and degraded over time as a result of colonisation, there is an opportunity to engage the *mātauraka Māori* that has persisted with another knowledge, such as soil science, to look back in time and affirm or rebuild aspects of *mātauraka Māori* to reconnect with the *whenua*. Where these knowledges interact, the strengths of both are harnessed and enhanced through mutual learning. Fig. 11 illustrates the research process as it has occurred in this study, following the form of the He Awa Whiria framework. The contributions of each knowledge stream are clearly

identifiable, along with the mutual learning that emerged from their interaction.





An example of the interaction between the knowledges occurred while attempting to identify what the soils had been fertilised with, as presented in Section 4.4. The potential for seaweed and penguin guano fertilisers initially considered was identified from other studies within the *takiwā* (Jacomb, 2000; Morris, 1994), with the identification of the seaweed source in one of

these studies coming from *mātauraka Māori* (Morris, 1994). The initial micronutrient analysis in the Pōhatu case study found that these were not likely to have been significant inputs to the *māra* soils. This finding was discussed with *Mana Whenua*, who suggested that *tī kouka* ash may have been a potential source of nutrients due to the association of its harvesting time being aligned with *kūmara* planting, as discussed by Payne (2020). *Tī kouka* is processed by cooking pounded roots and stems in large ovens, called *umu tī*, to produce *kauru*, a sugary substance important as a carbohydrate source (Best, 1976; Payne,

2020). It was proposed that the ash by-product from this process could have been applied to the *māra* as a fertiliser. The micronutrient analysis and PCA suggest that this is worth further investigation, with some parts of the tree reflecting the nutrients identified in the soil, with *kānuka* also being a likely input. Further examples of the interaction between knowledges are presented in Fig. 11.

The guidance the He Awa Whiria framework provides to researchers, knowledge holders, and stakeholders involved who are

external to at least one of the knowledge streams is valuable and ensures that each stream is constantly front of mind throughout the research process. The He Awa Whiria framework has facilitated the weaving of *mātaruaka Māori* and soil science in identifying past horticultural land use at Pōhatu, ensuring that knowledges are considered and applied with equal value without prioritising one over the other. In addition to the findings that support *mātauraka Māori* and the potential for *Mana Whenua* to rediscover and redevelop *māra* at this location, the process of applying the He Awa Whiria framework has provided several

key learnings and understandings when doing TDR and the value of extending the boundaries of soil science into a transdisciplinary space.

A fundamental aspect of TDR is engagement, which is reliant on establishing relationships built on mutual trust and understanding with stakeholders, Indigenous knowledge holders, and researchers from other disciplines (Robson-Williams et

al., 2023; Scholz, 2011; Stein et al., 2024). Relationships are a critical factor in working beyond the boundaries of soil science through TDR (Bouma et al., 2015; Keesstra et al., 2016). This involves fostering interdisciplinary collaborations to integrate insights from various scientific disciplines and facilitate non-academic engagement (Bennich et al., 2020; Hansson and Polk, 2018; Hirsch Hadorn et al., 2008; Lang et al., 2012). Additionally, embedding Indigenous lenses and engaging with diverse stakeholders can provide a more comprehensive understanding. The application of the He Awa Whiria framework in this study

has demonstrated how the knowledge of stakeholders can guide and inform soil science, and the value of utilising mutual learning opportunities through means of engagement that avoided structured interviews as extractive interactions and allowed for more open shared learning. Mutual learning cannot occur in isolation when the knowledges are applied and engaged with in isolation. This contrasts with how Western science is usually conducted, which often operates in silos, forming part of an important but disjointed picture (Gibbons et al., 1994; Lönngren and van Poeck, 2021). The benefits of research at the interface

between knowledges are only realised when relationships are genuine and strong. The Pōhatu case study has provided an





**Figure 11: Schematic application of the He Awa Whiria framework in this case study of understanding past food-landscapes at Pōhatu.**





example of this through ongoing engagement and discussion with *Mana Whenua* throughout the research process, building on the relationships established prior to the identification of the research questions.

Self-awareness is essential when developing meaningful relationships. Thus, positionality is a crucial element in engaging with a TDR methodology, despite being an uncommon practice in the bio-physical sciences. Positionality involves reflecting on the grounds on which one is engaging with the research and how they are prepared to approach the challenge (Hausermann and Adomako, 2022). This requires recognition and reflection of the researchers' biases, beliefs, cultural background, and experiences which can influence the research outcomes, as well as the research process (Bourke, 2014), which is particularly

important in TDR. By incorporating this self-awareness, TDR applications involving soil scientists can strive for greater objectivity and account for the potential influence of researchers' backgrounds on their interpretations and findings (Brown, 2023).

Soils tell a story of the environment and its history. Across most of the landscape, these stories are predominantly about the

climate, plants and animals, and the factors that resulted in its formation (Jenny, 1941). These stories can be read and understood by soil scientists and contribute to understanding the capabilities and limitations of the soil for production, conservation, and land use. In other places, these stories of the environment are interwoven with the stories of people present in these places and how they interact with their environment. To fully understand these stories, more than soil science is needed. While soil science can identify the words that make up the story, it is the relationships with people and their place-

based knowledge that give meaning to the properties of the soil, enabling the story to be told.

This case study and other local examples (for example, McFadgen and Adds, 2019) weave together knowledges to address research questions identified by *Mana Whenua*. Learnings from these applications can be applied when addressing global sustainability and soil security challenges in local contexts. In recognising the importance of place-based knowledge and

approaches (Saunders et al., 2023), opportunities to move beyond the boundaries of soil science can be realised. Learnings from the Pōhatu case study have supported the development of the recently proposed FLN framework, which aims to apply a holistic approach to understanding the (dis)connections between soil, food, and people in local, place-based contexts (Gillespie et al., 2024).

TDR is a different way of doing science. Knowledge production through applying a TDR approach is inherently complex due to its iterative and collaborative nature. It challenges the linear methods of Western-trained researchers, by moving beyond disciplinary boundaries and practices that characterise Mode 1 knowledge production (Gibbons et al., 1994). Our case study contributes to establishing a valuable foundation for understanding the interface of different knowledge systems, providing insights for other researchers looking to move beyond disciplinary boundaries.




A key part of this study has been to approach these knowledges with an open mind and consider the whole of the environment in understanding connections. Smith (2021), in discussing Cook's voyage to Aotearoa New Zealand, reflects that Banks' description of the people as using the "*same detached eye*" (p. 94) as used for their descriptions of the flora and fauna. In contrast, people and social aspects are considerations that cannot be separated from the environment in *mātauraka Māori*

(Hikuroa, 2017). By responding to the needs, questions, and priorities of local and Indigenous communities — alongside them —Western science begins to relinquish some of the power it has monopolised in research and investigation (Anthony-Stevens and Matsaw Jr, 2020; Lauter, 2023). This shift in power does not diminish the status of Western science. Rather, it elevates local and Indigenous knowledge and knowledge holders, recognising and valuing their ontological and epistemological foundations (Anthony-Stevens and Matsaw Jr, 2020; Chapman and Schott, 2020; Henri et al., 2021; Hill et al., 2020; McGregor

et al., 2010). Although this transition is confronting and is an uncomfortable process, it is necessary. As soil scientists, if we do not move beyond disciplinary boundaries, how can we truly address the complex challenges that society faces — and will continue to face?

## 6 Conclusion

By weaving together *mātauraka Māori* and soil science, understandings of past horticultural land use at Pōhatu have been

advanced. The findings of this study indicate that *kūmara* was grown where the soils have been modified with gravel additions. Soils were primarily fertilised with organic material from terrestrial plants, likely dominated by *kānuka*. It is evident from both *mātauraka Māori* and radiocarbon dating *kūmara* production occurred during Kāi Tāhu occupation of the bay, with refining assessment applying alternative methods required to determine if Kāti Mamoe were also growing *kūmara* here prior to the arrival of Kāi Tāhu. Furthermore, this study has provided an example of activating the He Awa Whiria framework. In the

Pōhatu case study, mutual learning at the intersection of knowledges occurred multiple times, with *mātauraka Māori* providing context or identifying next steps when understanding and interpreting the findings and results obtained using soil science methods. While this is only one of the many conceptual frameworks that have been developed, its application involving soil science provides guidance for TDR.

In this example, *mātauraka Māori* guided where to look and what to look for, and then soil science methods were applied to build on this and identify relevant details. Through ongoing engagement and relationship building, the findings were considered and interpreted in the context of *mātauraka Māori*. Interpreting the results by applying soil science only does not tell the complete story; however, when combined with *mātauraka Māori*, a more complete and meaningful understanding was achieved.




Soil science alone cannot sufficiently understand and address the complex global challenges rooted in soil security. This case study has demonstrated the value of weaving together soil science and Indigenous knowledge to understand the connections between soil and people in place-based contexts.


**Author contributions:**

JG: conceptualisation, fieldwork, methodology, data acquisition, data curation, formal analysis, data interpretation, visualisation, writing and editing (original draft and review and editing); MP: knowledge provision, conceptualisation, data interpretation, writing (reviewing); DP: relationship establishment, conceptualisation, methodology, data interpretation,

writing (reviewing and editing); SE: conceptualisation, methodology, writing and editing (original draft and review and editing); DJ: methodology, data interpretation, writing and editing (original draft and review and editing); CS: conceptualisation, methodology, data interpretation, writing (reviewing and editing); JC: conceptualisation, methodology, data interpretation, writing (reviewing and editing).

**Competing interests:** The authors declare that they have no conflict of interest.

**Acknowledgements:** This research was completed as part of a PhD at Te Whare Wanaka o Aoraki Lincoln University in Aotearoa New Zealand. It was funded by Manaaki Whenua Landcare Research and the Lincoln University Joint Postgraduate School scholarship. We also thank *Mana Whenua* of Koukourārata for the opportunity to work alongside you in the *māra*, the

Helps family for access to sample at Pōhatu, and the Pōhatu Penguins team for hosting us.



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
