# Peer review of "Research at the interface between Indigenous knowledge and soil science; weaving knowledges to understand horticultural land use in Aotearoa New Zealand"

_EGUsphere, 2024_

## Referee Comment (RC1)

Review of Gillespie et al. Research at the interface between Indigenous knowledge and soil science; weaving knowledges to understand horticultural land use in Aotearoa New Zealand (https://doi.org/10.5194/egusphere-2024-3546 Preprint).

(Note "L" refers to line number in the manuscript)

This manuscript (ms.) presents a transdisciplinary study of a traditional agricultural system in New Zealand, combining Māori indigenous knowledge with soil science. The ms. is well-written and organized, and contains important, interesting findings and views that are well within the scope of SOIL, and will help advance the interface between indigenous knowledge (IK) and ethnopedology. The present study can be published after some comments and questions relating to IK and traditional agriculture, and technical soils components, are addressed.

However, reflecting the authors' recognition of the need for more empirical studies to better quantitatively and scientifically support their transdisciplinary approach (TDR), I strongly encourage the authors to consider a more substantial scientific soil study going forward in their future continuation of this research. This would involve more robust sampling and data collection and analyses of traditional fields and soils, and comparison of the agricultural soils with reference (control) natural soils, and perhaps comparisons of a range of Māori field systems. A more quantitative (increased sample size and use of controls) would help to more rigorously test initial findings and inferences that the authors present in their current stage of research. This would not only help support Māori agriculture and food sovereignty development, but would also help advance soil health and food security internationally.

Main Comments and Questions

These comments mainly fall into two areas: 1) the authors' TDR approach regarding Māori traditional knowledge and goals regarding agriculture, and 2) technical aspects relating to the data collection, analysis, and interpretation of the agricultural fields, soils, phytoliths, charcoal/ash, and stable C and N isotopes.

1. The authors' TDR and collaborative research framework demonstrates how far ethnopedology studies have advanced from the early days: from recognizing the importance of indigenous knowledge in its own right, to now much more active engagement with indigenous communities in planning and conducting the research to incorporate those communities' desired outcomes. In this case, those outcomes involve traditional agriculture in relation to food sovereignty and food security. Another component the authors could address going forward is how to encourage community members, especially younger people, to obtain more education and training, and scientific expertise, so that they can more actively conduct, co-lead, and take the lead in the research, while also incorporating and conserving their heritage of traditional agricultural knowledge, language, and culture. Research areas for community members could be soil science, plant and crop sciences (e.g., phytolith and stable isotope analyses such as done in this study, and genetics), agricultural management, among others.

In terms of writing, the use of TDR language and the conceptual framework regarding the importance of indigenous knowledge and active participation and collaboration with indigenous communities is appropriate and understandable. The He Awa Whiria/two streams framework is useful, and Figure 11 is very good. However, the TDR wording can become excessive for scientists or the public unfamiliar with this approach, so please also keep in mind the need to

communicate clearly and concisely with simpler and non-repetitive language, depending on the audience. Also, generalizing about Western science as narrow (e.g., L593-4, 626) to promote TDR, can come across as unnecessary stereotyping.

2. The multiple kinds of data (soils, phytoliths, stable C and N isotopes, charcoal and ash) explored in this study are excellent, but an expanded sample design is essential as the studies hopefully go forward. The current data and findings are likely sufficient for this paper, but I encourage the authors to point to the future of this study, especially the need for an expanded and more robust sampling design, data collection, and subsequent analyses to test their current finding and inferences. Currently, it appears that only one specific field of earthen raised mound fields was sampled, whereas more sampling would allow for statistically valid comparisons to test current data hypotheses and inferences. With one field studied so far with two soil pits, more work is needed to test if current findings are representative. I realize that perhaps only one field was identified in the immediate study area, but maybe more could be found, or the soil study expanded to other areas where traditional fields have been identified, as indicated in L346-7 etc.

This research would also greatly benefit from incorporation of reference or control soil sampling; i.e., uncultivated/natural soils with similar comparable geomorphic and pedogenic settings, if these are available. This would help to test the anthropogenic soil change and phytolith and isotope signatures inferred by the authors. The lack of reference soils outside of the field systems for comparison raises questions about the validity of some of the current interpretations, and renders them more speculative. Also, scientific-based comparisons among raised mound fields of different age or settings, or different kinds of Māori traditional field systems would add information that would benefit the scientific scope and interpretations overall. A more quantitative use of control natural soils, and expanded sample design for fields and their soils, would support the need for more empirical studies rightly called for by the authors.

While the current findings are interesting and compelling, they also should be considered more preliminary in my view because of the minimal sampling and lack of control natural areas for comparison. The phytolith data about sweet potato (*kūmara*) is convincing, but it seems like some of the conclusions are less certain than currently conveyed, especially those regarding the soils and their management (e.g., L529-30, 538-9, 553-4, 658-9). I think some of the conclusions should be less bold and more toned down, and future work needed to test initial findings based on few samples should be acknowledged more.

Other specific comments and questions
The incorporation of Māori terms is important, and most of these terms are reasonably defined when first stated. However, I suggest that the authors add a table of Māori terms (at least the key ones used more than once) to refer to easily – it may be challenging for many readers to remember the word meanings when going through the whole paper, so being able to access the terms in one place (i.e., table or mini-glossary) would be helpful.

State more explicitly in the Methodology section (e.g., first paragraph of Section 3.2 starting with L275, and maybe around L306 and beginning of Findings L314) that you were sampling an inferred traditional earthen raised mound field system. A little more than just stating features as "mounds" or "earth rows" would be clearer to readers.

Questions about Tables 1 and 2, and Figure 5, and need to give more information:
Soil morphology and horizon designations (some symbols and terms in the New Zealand system may not be familiar to all readers): color (all moist colors? What are the $2^{nd}$ colors given in parentheses?); texture (explain the abbreviations, especially "Z"); structure (is polyhedral same as granular or ?); Size (state in column label or caption that this is coarse fragment size; does % abundance mean volume %?); horizon designations in Tables and Fig. 5: does A/B mean same as AB or discrete A and B parts within the horizon?); meaning of (f) and (g) in parentheses for Bw? In Tables, add a column with the specific depth intervals for each horizon. For Figure 5, state the scale units (e.g., numbers are 10 cm intervals). Also, I am not seeing the arrows for "beach gravel additions" stated in the Figure 5 caption. You indicate an "Ap2" in the Figure 5b caption, but that's not shown in the photo – did you mean "2Ap"?

Regarding the landslide: is this a natural landslide or is there possible anthropogenic influence from the agriculture – e.g., could the field construction and use have induced the landslide? Is this landslide an isolated case, or are these landslides common. Need more context here, and this also shows the need to sample more fields.

Good multiple analyses in soil chemistry (especially Mn in Figure 9) and C and N isotopes in relation to possible soil amendment inputs for nutrient management such as penguin guano and seaweed. Again, all of these analyses (soil chemistry, stable isotopes, charcoal/ash, gravels) and interpretations regarding soil modification would benefit from comparison with some kind of baseline data from control (nonagricultural soils that match the agricultural soils in natural pedogenesis and ecological and geomorphic setting), if they are available. With Mn for example, you indicate increases in inferred modified horizons but just for Pit 2 (what about Pit 1?). Incorporating more fields for soils analyses, and comparison with surface horizons etc. in natural soils, could help better characterize Mn distribution, variability, and test whether Mn is diagnostic of amendment inputs. The greywacke gravel input inference in relation to IK seems valid, but greater sample size and comparison with similar natural horizons in control soils could allow you to be more definitive and certain that this gravel could only be from deliberate input for management (e.g., are you certain that the geologic occurrence and distribution of greywacke isn't more complex?). Monitoring natural control soils along with the agricultural soils could also allow you to test and quantify drainage and soil warming benefits of gravel inputs.

L404 – explain a bit more about manure. Are you saying that use of manure is totally prohibited by Māori?

L621: define FLN in this ms. (Food-Landscape Network).

Why isn't "ethnopedology" mentioned in the text (only indirectly in one reference). Topics covered in this ms. seem closely related to the subdiscipline of ethnopedology, and seems like it should be mentioned if not highlighted.

References: good citation of literature on traditional Māori agriculture. It might also be useful to also consider citing some literature on traditional agricultural systems and soils research elsewhere in Polynesia and Oceania that may be relevant to your study and findings. Examples:

Lincoln, N. et al. (2014). Indicators of soil fertility and opportunities for precontact agriculture in Kona, Hawai'i. *Ecosphere, 5*(4), art42. doi:10.1890/es13-00328.1

Ladefoged, T. N. et al. (2018). Soil nutrients and pre-European contact agriculture in the leeward Kohala field system, Island of Hawai'i. *Archaeology in Oceania, 53*(1), 28-40. doi:10.1002/arco.5138

Sherwood, S. C. et al. (2019). New excavations in Easter Island's statue quarry: Soil fertility, site formation and chronology. *Journal of Archaeological Science, 111*, 104994. doi: https://doi.org/10.1016/j.jas.2019.104994

Autufuga, D. et al. (2023). Distribution of soil nutrients and ancient agriculture on young volcanic soils of Ta'ū, American Samoa. *Soil Systems, 7*(2), 52. doi:10.3390/soilsystems7020052

Also consider other stable isotope research on possible guano use in ancient agriculture: Santana-Sagredo, F. et al. (2021). White Gold in the Atacama Desert: Isotopic evidence for early seabird guano use in South America. *Nature Plants, 7*, 152-158. https://doi.org/10.1038/s41477-020-00835-4

Szpak, P. et al. (2012). Stable Isotope Biogeochemistry of Seabird Guano Fertilization: Results from Growth Chamber Studies with Maize (Zea Mays). *PLOS ONE, 7*(3). doi:10.1371/journal.pone.0033741

---

## Author Response (AR1)

| | Comment | Address |
|---|---|---|
| 1 | The TDR wording can become excessive for scientists or the public unfamiliar with this approach, so please also keep in mind the need to communicate clearly and concisely with simpler and non-repetitive language, depending on the audience. | L627 – Thank you for this suggestion. Where possible excess language (E.g. Mode 1 knowledge production replaced with Western scientific knowledge production in line 659) has been omitted In the section discussing the themes present in the transdisciplinary research literature (L56+), the TDR acronym has been removed from the bullet points to reduce repetition |
| 2 | Generalisation of Western Science as narrow | Line 658 modified to reflect that this is often, but not always the case. L626 in its original form uses the terminology 'usually' and 'often' indicating that the are other ways that Western Science can operate |
| 3 | Expansion of sampling for the future of this study, point to the future | This has been addressed at the end of Section 4 (L541). |
| 4 | One field of study used – could more be found or study expanded to other locations | Addressed with point 3 (L 541). More sites within the bay were looked at in less detail, and while some inference may be drawn from this, to prevent what is already a long ms becoming even longer these findings are not discussed. |
| 5 | Reference/control soil sampling – uncultivated/natural soil with similar comparable geomorphic and pedogenic settings, if these are available. This would help to test the anthropogenic soil change and phytolith and isotope signatures inferred by the authors. The lack of reference soils outside of the field systems for comparison raises questions about the validity of some of the current interpretations, and renders them more speculative. Also, scientific-based comparisons among raised mound fields of different age or settings, or different kinds of Māori traditional field systems would add information that would benefit the scientific scope and interpretations overall. A more quantitative use of control natural soils, and expanded sample design for fields and their soils, would support the need for more empirical studies rightly called for by the authors | This is an interesting thought and an approach that I have come across in the local literature. In relation to some of the later comments (greywacke and soil temperature, soil fertility), I can see that this would be a useful avenue for further research. |

| | | |
|---|---|---|
| 6 | While the current findings are interesting and compelling, they also should be considered more preliminary in my view because of the minimal sampling and lack of control natural areas for comparison. | As for point 3 - in L541 |
| 7 | The phytolith data about sweet potato (kūmara) is convincing, but it seems like some of the conclusions are less certain than currently conveyed, especially those regarding the soils and their management (e.g., L529-30, 538-9, 553-4, 658-9). I think some of the conclusions should be less bold and more toned down, and future work needed to test initial findings based on few samples should be acknowledged more. | These have been addressed, acknowledging the opportunity to look into this further |
| 8 | Addition of a table of Māori terms | This has been added to the end of the manuscript |
| 9 | State more explicitly in the Methodology section (e.g., first paragraph of Section 3.2 starting with L275, and maybe around L306 and beginning of Findings L314) that you were sampling an inferred traditional earthen raised mound field system. A little more than just stating features as "mounds" or "earth rows" would be clearer to readers. | Thank you for this suggestion, this change has been made. |
| 10 | Questions about Tables 1 and 2, and Figure 5, and need to give more information: Soil morphology and horizon designations (some symbols and terms in the New Zealand system may not be familiar to all readers): color (all moist colors? What are the 2nd colors given in parentheses?); texture (explain the abbreviations, especially "Z"); structure (is polyhedral same as granular or ?); Size (state in column label or caption that this is coarse fragment size; does % abundance mean volume %?); horizon designations in Tables and Fig. 5: does A/B mean same as AB or discrete A and B parts within the horizon?); meaning of (f) and (g) in parentheses for Bw? In Tables, add a column with the specific depth intervals for each horizon. | Thank you for highlighting this
- Morphology and horizon designation definitions have been provided
- Moist colour has been defined in the column label
- Colour in parentheses identified as mottle colour
- Texture abbreviations are provided in the table caption
- Structure definitions provided
- Coarse fragment size, % abundance is clarified in the table
- Depth interval column added |
| 11 | For Figure 5, state the scale units (e.g., numbers are 10 cm intervals). Also, I am not seeing the arrows for "beach gravel | - Scale unit added (10 cm intervals)
- Arrows added |

| | | |
|---|---|---|
| | additions" stated in the Figure 5 caption. You indicate an "Ap2" in the Figure 5b caption, but that's not shown in the photo – did you mean "2Ap"? | - Ap2 in caption corrected to 2Ap |
| 12 | Regarding the landslide: is this a natural landslide or is there possible anthropogenic influence from the agriculture – e.g., could the field construction and use have induced the landslide? Is this landslide an isolated case, or are these landslides common. Need more context here, and this also shows the need to sample more fields. | Further discussion is provided. These landslides are common, particularly after storm events, where the poorly structured, unstable Pallic Soils slip. This occurs in both areas with dense vegetation coverage, as well as open areas that have been cleared of their original forest cover |
| 13 | Again, all of these analyses (soil chemistry, stable isotopes, charcoal/ash, gravels) and interpretations regarding soil modification would benefit from comparison with some kind of baseline data from control (nonagricultural soils that match the agricultural soils in natural pedogenesis and ecological and geomorphic setting), if they are available. | See response to comment 5 |
| 14 | With Mn for example, you indicate increases in inferred modified horizons but just for Pit 2 (what about Pit 1?). Incorporating more fields for soils analyses, and comparison with surface horizons etc. in natural soils, could help better characterize Mn distribution, variability, and test whether Mn is diagnostic of amendment inputs. | The figure now shows pit 1 also. Pit 2 shows the differences particularly clearly due to the burying slowing the original modified horizon's development, while development has continued in the modified horizons in pit 1, as reflected in Figure 5.

As you state, a more intensive study across the area would provide further understanding and characterisation. |
| 15 | The greywacke gravel input inference in relation to IK seems valid, but greater sample size and comparison with similar natural horizons in control soils could allow you to be more definitive and certain that this gravel could only be from deliberate input for management (e.g., are you certain that the geologic occurrence and distribution of greywacke isn't more complex?). | It is not possible for the greywacke to have been emplaced in this location by any natural means. There are no streams/creeks in the immediate vicinity of the field that could have carried them here, even in flood events, with the parent material that would have been transported if they were present being basalt anyway. The aspect of the slope and its elevation above sea level would prohibit this from being a tsunami deposit, with other tsunami indicators being absent.
Looking at this field specifically, augering occurred across the slope, both on and between rows, with an absence of gravels present between the earthen |

| | | rows. This detail has been added at line 338 |
|---|---|---|
| 16 | Monitoring natural control soils along with the agricultural soils could also allow you to test and quantify drainage and soil warming benefits of gravel inputs. | Monitoring temperature of natural soils alongside modified soils would be an interesting study to undertake at multiple different sites (across Aotearoa New Zealand) where this type of management practice has been applied. This is something to look to for future research. |
| 17 | L404 – explain a bit more about manure. Are you saying that use of manure is totally prohibited by Māori? | Traditionally, manures were not used in order to prevent illness. Some accidental/incidental incorporation of guano may have occurred, but would not have been deliberately added. It is likely that the other site nearby (Morris, 1994) the 'natural' fertility of these soils by the penguins was utilised, but further additions as seen in other cultures (as discussed in the suggested references), would not have occurred. This section has been ammended for clarity |
| 18 | L621: define FLN in this ms. (Food-Landscape Network) | Full version written |
| 19 | Why isn't "ethnopedology" mentioned in the text (only indirectly in one reference). Topics covered in this ms. seem closely related to the subdiscipline of ethnopedology, and seems like it should be mentioned if not highlighted | This is a good point, thank you for highlighting it. This has been included in lines 55, 119 and 198 |
| | Suggested references | Thank you for these suggestions. While the others focused on different indicators than what we have looked at, they would be useful in a more comprehensive, and wide spread study, as discussed previously. |

Reviewer 2

| | Comment | Response |
|---|---|---|
| 1 | Addition of a glossary of Māori terms | Added at end of ms |
| 2 | Who are the individuals guiding the explication of Māori texts, and who is not involved? | Co-author of this paper MP has been the key guide in this area and is the mātauraka Māori/knowledge holder for the study site and community. We engaged with relevant local knowledge needed to progress the research through MP, as we explain in the 'Engagement |

| | | with Mana Whenua' section (section 3.1) in the methods. In this section we also describe how the relationship with the broader community at Pōhatu was also established through planting activities (see lines 264-276). While we acknowledge that it could be interpreted that MP acted as a spokesperson for his hapū/sub-tribe, in line with tikanga Māori/ Māori customs. |
|---|---|---|
| 3 | Who holds the texts and access to them? | As noted above, with adjusted explanation in section 3.1, co-author of this paper, MP, is the holder of this mātauraka Māori for the hapū/sub-tribe, mentioned in line 268-269. Importantly, this knowledge is not in 'text' format - it is oral knowledge that is passed from generation to generation. Lines 108-110 have been expanded to reflect this. In this research, this oral knowledge came in the form of pūrakau/stories, waiata/songs and ingoa wahi/names (see section 1.1 and section 3.1 for specific details of this knowledge). Furthermore, the holder of Mātauraka Māori does not act as a gatekeeper, preventing access to this knowledge - but according to cultural protocols it was vital to engage directly with MP to access this knowledge as explained in comment 1 above. |
| 4 | The authors mention awareness of power dynamics between Māori and non-Māori team-members, but I would like to understand more about how the team negotiated the Māori side, who chose which texts and why? | Our mention of power dynamics in line 255 refers to the power imbalance between Western science disciplines and Mātauraka Māori, not members of the research team. The guiding principle of this research was to counter this imbalance, hence our use of the He Awa Whiria/Braided Rivers framework to enable each knowledge stream to exist independently and also make an equal contribution to the research. As a research team, we built genuine and nurturing relationships with each other through the course of this research - this was particular the case for the lead author, JG, who lead relationship building with mana whenua/the local Indigenous community (see section 3.1) with a large degree of humility (mentioned on line 253). As explained in comments 1 and 2 above, MP directed us in our engagement |

| | | with relevant mātauraka Māori/knowledge in the form of pūrakau/stories, waiata/songs and ingoa wahi/names. |
|---|---|---|
| 5 | Additionally, is there a gender dimension in any texts? | This is an interesting consideration but in this research we were not looking for or at gendered dimensions of mātauraka Māori, and we did not encounter any findings of relevance in the course of this research. |
| 6 | The conclusions of this article focus more on the process of doing the TDR research than on the outcomes as these relate to why this research matters for soil and food security. A stronger article would come back to the open question of addressing the soil and food security challenges, and why this weaving of knowledges is better to address these challenges than the separate knowledges. | The final paragraph of the paper reflects this, and has been further expanded to reflect the 'why' question. |
| 7 | I would also like the authors to consider how the specific findings of their case study matter beyond this particular location. Inherent in any work trying to weave together TEK and western science is the tension of exceptionalism vs. generalizability, and I would like to see this addressed in the conclusions. | This is introduced in L651 (discussion), and reflected again in the conclusion |
| 8 | National Science Foundation (NSF) Advisory Committee for Environmental Research and Education (AC-ERE). 2022. Engaged Research for Environmental Grand Challenges: Accelerating Discovery and Innovation for Societal Impacts. National Science Foundation: Alexandria, VA. | Thank you for this reference. This is a valuable read that aligns with our motivations. |

Reviewer 3:

Thank you for your support of our manuscript, it is pleasing to see the key themes we aimed to articulate reflected in your response.

To assist with engaging with the Māori language aspects of the manuscript we have provided a glossary at the end of the document.

Editor:

Thank you for your support of our manuscript. The etics/emics concepts have been introduced in line 54, linking it to TDR.